# End-to-End Single-Channel Speaker-Turn Aware Conversational Speech Translation

**Juan Zuluaga-Gomez**[*†], **Zhaocheng Huang**[**‡], **Xing Niu**[**‡], **Rohit Paturi**[‡],
**Sundararajan Srinavasan**[‡], **Prashant Mathur**[‡], **Brian Thompson**[‡], **Marcello Federico**[‡]
[†]Idiap Research Institute & EPFL          [‡]AWS AI Labs

juan.zuluaga@eu4m.eu
{davidhzc, xingniu, paturi, sundarsr, pramathu, brianjt, marcfede}@amazon.com

## Abstract

Conventional speech-to-text translation (ST) systems are trained on single-speaker utterances, and they may not generalize to real-life scenarios where the audio contains conversations by multiple speakers. In this paper, we tackle single-channel multi-speaker conversational ST with an end-to-end and multi-task training model, named Speaker-Turn Aware Conversational Speech Translation, that combines automatic speech recognition, speech translation and speaker turn detection using special tokens in a serialized labeling format. We run experiments on the Fisher-CALLHOME corpus, which we adapted by merging the two single-speaker channels into one multi-speaker channel, thus representing the more realistic and challenging scenario with multi-speaker turns and cross-talk. Experimental results across single- and multi-speaker conditions and against conventional ST systems, show that our model outperforms the reference systems on the multi-speaker condition, while attaining comparable performance on the single-speaker condition. We release scripts for data processing and model training.[1]

## 1 Introduction

Speech translation (ST) has seen wide adoption in commercial products and the research community (Anastasopoulos et al., 2021, 2022) due to its effectiveness in bridging language barriers. ST aims to translate audio of source languages into text of the target languages. This problem was tackled by a cascaded approach that pipelines Automatic Speech Recognition (ASR) and Machine Translation (MT) over the last few decades (Waibel et al., 1991; Vidal, 1997; Casacuberta et al., 2008, *inter alia).* However, end-to-end speech translation (E2E-ST) systems (Berard et al., 2016; Weiss et al.,

---

[*]Work conducted during an internship at Amazon.
[**]Corresponding authors with equal contributions.
[1]https://github.com/amazon-science/stac-speech-translation

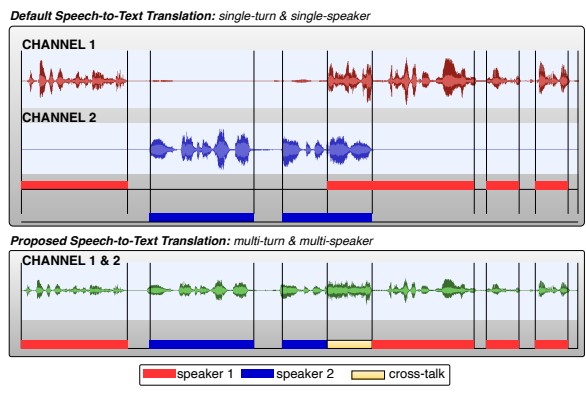

Figure 1: A two-speaker multi-turn conversational segment. Previous work focuses on separated channels without considering cross-talks and speaker-turns (top). STAC-ST targets a more challenging scenario where multiple speakers converse with occasional cross-talks due to merged channels (bottom).

2017, *inter alia)* have recently gained increasing interest and popularity thanks to their simple architecture, less error propagation (Etchegoyhen et al., 2022), efficient training process, and competitive performance (Inaguma et al., 2019).

Despite significant recent advances in E2E-ST (Gheini et al., 2023; Wang et al., 2023), most ST systems to date have focused on translating isolated speech utterances from monologue speech (Di Gangi et al., 2019), read speech (Kocabiyikoglu et al., 2018) or prompted speech (Wang et al., 2021). Being trained on single-turn utterances, these systems may lack the ability to handle real-life scenarios in which multiple speakers converse, and sometime overlap, in the same audio channel (Post et al., 2013).

In this work, we tackle the more challenging task of multi-speaker conversational ST. We refer to it as *multi-turn & multi-speaker* (MT-MS), as opposed to single-turn, which most ST systems implicitly assume. This is illustrated in Figure 1, where a "conversation" between two speakers recorded with separate channels (top) becomes more difficult to

translate if the channels are merged (bottom), due to the introduction of speaker-turns and cross-talks. In particular, ST with cross-talks and speaker-turns is difficult because speech content of different sentences is mixed up or switched. While MT-MS speech has been studied in ASR (Raj et al., 2022), to the best of our knowledge, this is the first paper that investigates it in end-to-end ST. We tackle MT-MS ST with an approach we named **S**peaker-**T**urn **A**ware **C**onversational **S**peech **T**ranslation (STAC-ST). STAC-ST is a multi-task training framework that combines ASR, ST and speaker-turn detection using special tokens in a serialized labeling format. It is inspired by a recent speech foundation model, Whisper (Radford et al., 2023), which jointly trains ASR, X-to-English ST, voice activity detection, and language identification with 680k hours of speech data using labeling-based multi-task learning. Our contributions are as follows:

1. We introduce the task of multi-turn & multi-speaker ST, including cross-talks and speaker-turns, that expands the realm of ST which has been limited to single-speaker utterances.
2. We propose an end-to-end model (STAC-ST) which achieves state-of-the-art BLEU scores on Fisher-CALLHOME, a corpus that allows to target MT-MS without degradation on single-turn ST.
3. We explore a zero-shot scenario where MT-MS ST data is not available for training. We show that STAC-ST improves ST up to 8 BLEU by leveraging MT-MS ASR targets, mitigating the necessity of parallel data, which is lacking within the community.
4. Besides serializing transcripts and translations at cross-talks, the STAC-ST model is also shown to learn the task of time-aligned speaker change detection.
5. We conduct extensive ablation studies on important aspects of STAC-ST, including joint modeling of ASR & ST, impact of model size (up to 300M parameters), data size, and integration of task tokens. Thus, we shed light on the best practices for building conversational MT-MS ST systems.

## 2 Related Work

**Joint ST & ASR Modeling** Recent works in ST have leveraged ASR training data to improve translation quality. In principle, joint ASR and ST modeling (Gheini et al., 2023; Soky et al., 2022)

requires 3-way parallel data for each training example, i.e., audio, transcript, and translation, as can be found, in limited amount, in the CoVoST (Wang et al., 2020, 2021) and MuST-C (Di Gangi et al., 2019) corpora. Prior work proposed to overcome the 3-way parallel data bottleneck by pseudo-labeling ST data (Gheini et al., 2023), or by pre-training an ASR model (van den Oord et al., 2018) on large multilingual data (Bapna et al., 2022; Zhang et al., 2023b) before training the joint ASR & ST model (Babu et al., 2022). Recently, the Whisper model (Radford et al., 2023) introduced an effective annotation format for jointly training ASR & ST with independent targets.

**Conversational Speech Translation** Work on conversational ST (Kumar et al., 2014b,a; Zanon Boito et al., 2022) has mainly focused on single-speaker speech, either segmented manually or automatically, via voice activity detection. Manual segmentation was assumed in recent studies, based on the Fisher and CALLHOME corpora, on cascaded ST (Kumar et al., 2014b), E2E-ST (Weiss et al., 2017; Peng et al., 2023), simultaneous ASR & ST (Soky et al., 2022), streamed ST (Deng et al., 2022), and multilingual ST (Inaguma et al., 2019). Automatic segmentation was instead deployed with the MSLT corpus (Federmann and Lewis, 2016) to target streamed ST (Xue et al., 2022) as well as language-agnostic streamed ST (Wang et al., 2023).

In this work, we report results on the Fisher-CALLHOME corpus (Post et al., 2013) which, similarly to the MSLT corpus, offers the opportunity to run contrasting experiments of single-speaker ST versus MT-MS ST, both without reference segmentation.

**Speaker-Turn and Cross-Talk in ASR** Speaker-turns and cross-talks have been explored in the ASR field and commonly termed, multi-talker ASR. Kanda et al. (2020) proposed a serialized output training (SOT) strategy for multi-speaker overlapped speech recognition with special tokens. At inference time, word and speaker tags are output in a serialized manner for an unlimited number of speakers. SOT was later ported to the streaming scenario (Kanda et al., 2022). However, SOT may produce frequent speaker changes, which can degrade the overall performance. Thus, Liang et al. (2023) proposed to explicitly incorporate boundary knowledge with a separate block for speaker change detection task and boundary constraint loss.

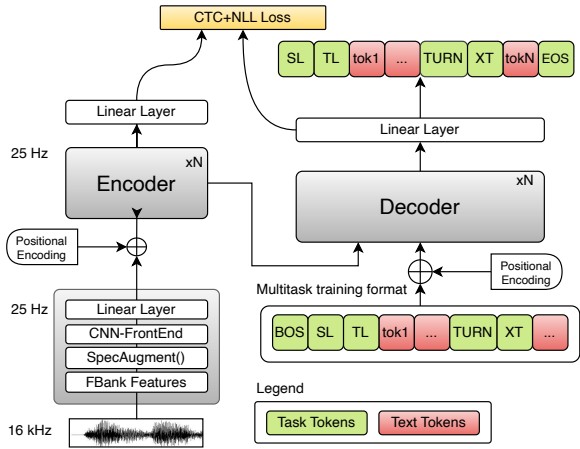

Figure 2: Proposed model architecture of STAC-ST for multi-turn & multi-speaker ST.

Multi-talker ASR has also been explored in the non-streaming (Huang et al., 2023) and streaming (Raj et al., 2022) setups. Multi-turn ASR has been explored in automatic dubbing (Virkar et al., 2023) of scripted content, a challenging case due to the high number of speakers and short segments (Brannon et al., 2023), but improvements have come from aligning (Thompson and Koehn, 2019, 2020) automatic transcripts with available production scripts. Another branch of research targets cross-talk & multi-talker ASR (Yang et al., 2023) using speech separation of long-form conversational speech (Paturi et al., 2022) but these techniques have difficulty handling variable number of speakers and are not optimized end-to-end for ASR improvements. However, how to effectively deal with multi-speaker conversational ST has been neglected.

## 3 Speaker-Turn Aware Conversational Speech Translation (STAC-ST)

This section describes our end-to-end multi-task learning model for multi-turn multi-speaker conversational ST.

### 3.1 System Diagram

Figure 2 illustrates the proposed STAC-ST multi-task learning framework for MT-MS ST. The model is an encoder-decoder Transformer architecture inspired by Vaswani et al. (2017). The multitask training format using special tokens (§3.2) was inspired by Whisper (Radford et al., 2023), while the integration of Connectionist Temporal Classification (CTC) loss (§3.3) was inspired by Watanabe et al. (2017).

STAC-ST has a standard front-end module. First,

frame-level 80-dimensional filterbank features are extracted from the audio[2] every 40ms. Second, we apply SpecAugment (Park et al., 2019) on the input audio features, an effective data augmentation technique that masks out certain regions of the input filterbank features. Then, the audio augmented features are passed to a 2-layer CNN that outputs a 5120-dim vector (flattened 2D→1D output tensor from the CNN layer). Finally, this vector feeds a linear layer that generates the input to the encoder model. The decoder takes the encoder outputs and generates a sequence of text. Formally, for each speech segment, the filterbank features can be represented as: $X = \{\mathbf{x}_t \in \mathbb{R}^F\}_{t=1}^T$ and the reference transcription or translation as: $Y = \{w_n \in V\}_{n=1}^N$. Where, $F$ is the feature dimension, $T$ is the number of speech frames, $N$ is the number of text tokens, and $V$ is the vocabulary. During training of STAC-ST, we concatenate independent datasets $D_{ASR} = (X, Y_{ASR})$ and $D_{ST} = (X, Y_{ST})$, for ASR & ST, respectively. Samples of training mini-batches are jointly drawn from $D_{ASR}$ and $D_{ST}$.

### 3.2 Serialized Labeling Based on Task Tokens

A key component of the model is the serialized multi-task labeling framework based on special tokens. As shown in Figure 2, besides the text tokens, special tokens are used to specify the task. There are four types of task tokens, i.e., [SL] (source language), [TL] (target language), [TURN] (speaker-turn), and [XT] (cross-talk).

The first two tokens are language tokens that define the task for either ST (when [SL] ≠ [TL]) or ASR (when [SL] = [TL]). At training time, we instantiate language tokens and prepend them to each sample of $D_{ST}$ and $D_{ASR}$, such as

```
ST: [ES] [EN] utterance translation.
ASR: [ES] [ES] transcripción de enunciados.
```

At inference time, both language tokens are preset to specify the desired task.

[TURN] and [XT] specify the auxiliary tasks of detecting speaker-turn changes and cross-talks, which are critical for MT-MS speech processing and more aligned to acoustic tasks. Note that cross-talks always occur during speaker-turn changes, so [XT] always follows [TURN].

We concatenate transcripts or translations sequentially, inserting [TURN] and [XT] tokens when needed. If utterances $u_t$ and $u_{t+1}$ overlap in time, we append the targets of utterance $u_{t+1}$ after utter-

---

[2]The audio is always down- or up-sampled to 16 kHz.

ance $u_t$. The order of utterances is determined by their start time. A demonstration of such serialization is shown below:

```
CHANNEL 1: |WORD1|        |WORD2 WORD3 ...|
CHANNEL 2:        |word1 word2|
Serialization: WORD1 [TURN] word1 word2 [TURN]
    [XT] WORD2 WORD3 ...
```

### 3.3 Joint CTC and NLL Loss

STAC-ST jointly models ASR and ST by balancing CTC (Graves et al., 2006) and Negative Log-Likelihood (NLL) losses (Chan et al., 2016), according to:

$$\mathcal{L} = \lambda \cdot \mathcal{L}_{CTC}(Y|X) + (1-\lambda) \cdot \mathcal{L}_{NLL}(Y|X), \quad (1)$$

$\mathcal{L}_{CTC}$ and $\mathcal{L}_{NLL}$ are computed by appending linear layers with dimension $V$ on top of the encoder and decoder, respectively. Figure 2 shows the proposed joint CTC/NLL loss training scheme (Watanabe et al., 2017). In practice, the CTC loss models a probabilistic distribution by marginalizing over all possible mappings between the input (audio features, sampled at 40 ms) and output sequence (transcription or translation). We refer readers to the original implementation by Graves et al. (2006), for more details. Moreover, CTC loss has been proven to aid ST by helping to stabilize encoder representations at early stages of training, i.e., allowing the decoder to learn soft alignment patterns faster (Yan et al., 2023). Note that we do not include language tokens, [SL] and [TL], for $\mathcal{L}_{CTC}$ computation because they do not correspond to acoustic features. Following previous work (Zhang et al., 2022, 2023a), we set the weight $\lambda$ of the CTC loss to 0.3.

## 4 Experimental Setup

This section introduces the datasets and metrics we used for evaluation, as well as architecture and training details of STAC-ST.

### 4.1 Conversational Multi-Turn & Multi-Speaker ST

We use the Fisher and CALLHOME corpora which respectively comprises 186 hr and 20 hr of audio and transcripts of telephone conversations in Spanish.[3] The Spanish-to-English translations are available from Post et al. (2013). We refer to them as Fisher-CALLHOME and summarize the data

[3]LDC2010S01, LDC2010T04, LDC96S35, LDC96T17

| Statistics | Fisher | | | | CALLHOME | | |
|---|---|---|---|---|---|---|---|
| | train | dev | dev2 | test | train | dev | test |
| Single-Turn Duration [hr] | 172 | 4.6 | 4.7 | 4.5 | 14.7 | 3.8 | 1.8 |
| Single-Turn #Utterance [k] | 139 | 4.0 | 4.0 | 3.6 | 15 | 4.0 | 1.8 |
| MT-MS Duration [hr] | 155 | 4.1 | 4.1 | 4.1 | 13.8 | 3.5 | 1.7 |
| MT-MS #Utterance | 22k | 572 | 580 | 583 | 1.9k | 482 | 242 |
| Speech activity [%] | 97 | 97 | 98 | 98 | 78 | 80 | 58 |
| Overlap ratio [%] | 12.7 | 14.5 | 16.8 | 11.2 | 11.7 | 14.6 | 11.8 |

Table 1: Fisher-CALLHOME corpus statistics.

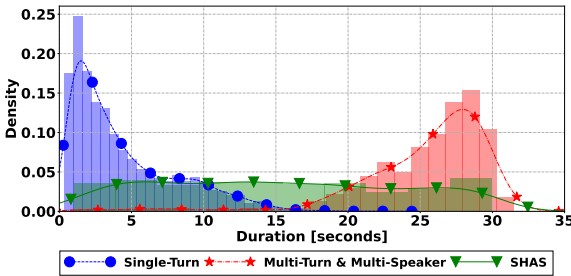

Figure 3: Fisher-CALLHOME test set distribution of segment length with three different segmentation approaches: single-turn, MT-MS, and SHAS.

statistics in Table 1. This corpus is well suited for MT-MS ST, as it contains a significant amount of labeled data and non-segmented (audio) long conversation between speakers. We merged Fisher and CALLHOME for training and up-sampled the audio to 16 kHz.

**Segmentation.** Each conversation on Fisher-CALLHOME occurred between two speakers with multiple turns over two channels (one speaker per channel). For MT-MS ST experiments, we merge the two channels into one, which creates natural speaker changes and cross-talks as illustrated in Figure 1. Human annotations in Fisher-CALLHOME provide time-aligned audio utterances, transcripts and translations, and have been used to segment each channel into single-turn utterances in prior work (e.g., Inaguma et al., 2019). Figure 3 plots the distributions of segment duration in the corpus. We observe that the majority of single-turn segments are less than 5 seconds long. To build models with manageable size and computation, following Radford et al. (2023), we segment the merged-channel conversations into chunks of up to 30 seconds. For this step, we first used an off-the-shelf VAD-based segmentation tool, SHAS (Tsiamas et al., 2022), but we realized that the resulting duration histogram is almost uniform and far from the natural segmentation. Hence, we decided to rely on the manual time annotations as follows. Starting from the first utterance $start$, we

find the farthest utterance $end$ such that $end - start$ is up to 30 seconds. We extract audio within this span as one segment and repeat this procedure until the last utterance $end$ is reached. Note that one segment may stretch over multiple utterance $start$ and $end$, so it may include silences, noise, speaker changes and cross-talks. We use this as the primary MT-MS segmentation strategy for both training and test data throughout the paper unless otherwise stated. More discussions can be found in Section 5.3.1.

## 4.2 Additional ASR & ST Corpora

Fisher-CALLHOME has limited training data size, so we explore additional corpora to improve our model and to evaluate its generalization ability. We also use the official CoVoST 2 (Wang et al., 2021) splits for Spanish-English ST (156 hr) and Common Voice[4] (CV, Ardila et al., 2020) splits for Spanish ASR (458 hr) as additional training data. Even though these corpora are not in the conversation domain, they may still help speech modeling in general.

CoVoST 2 and CV corpora are composed of single-turn pre-segmented utterances. To generate data consistent with our MT-MS segmentation, we randomly concatenate audio utterances and yield segments of up to 30 seconds. Note that these synthetic MT-MS segments contain no silences and cross-talks, but still have speaker-turn changes (labeled by [TURN]).

## 4.3 Evaluation Metrics

We report case-insensitive BLEU using Sacre-BLEU[5] (Post, 2018) for translation and Word Error Rate (WER) for ASR. Note that we (1) remove all special task tokens before computing each metric and (2) evaluate on MT-MS segmentation unless otherwise stated.

## 4.4 Hyper-Parameters

We experiment with three model sizes, S(mall), M(edium), and L(arge), with increasing dimension (256, 512, 1024), number of encoder layers (12, 14, 16), number of heads (4, 8, 16), with same number of decoder layers (6) and FFN dimension set to 4x the model dimension. Their numbers of parameters are 21M, 86M, and 298M, respectively. We use the S-size model by default and scale up to

larger sizes when out-of-domain training data are added. We apply BPE sub-words (Sennrich et al., 2016) on both translations and transcripts with 5K operations. We create a joint BPE model for the language pair or when we add CV+CoVoST2 corpora (only §5.3.2 and §5.3.3).

We train for 100k steps the S-size models and 200k steps the M- and L-size models. We use AdamW (Kingma and Ba, 2015) optimizer with a peak learning rate of $5e^{-3}$ for the S model and $1e^{-3}$ for M and L models. The learning rate scheduler has warmup and cooldown phases, both taking 10% of the total training steps (Zhai et al., 2022). We set dropout (Srivastava et al., 2014) to 0.1 for the attention and hidden layers, and use GELU (Gaussian Error Linear Units) as the activation function (Hendrycks and Gimpel, 2016). We use gradient norm clipping (Pascanu et al., 2013)[6] and SpecAugment (Park et al., 2019) for data augmentation. The training configuration and architecture are based on a LibriSpeech recipe for Transformer-based ASR from the SpeechBrain toolkit (Ravanelli et al., 2021).[7]

## 5 Results

Our experimental results document three properties of the STAC-ST model: (1) robustness to the MT-MS ST condition with no degradation in the single-turn ST condition; (2) ability to leverage speaker-turn and cross-talk information, which translates into improved WER and BLEU scores; (3) ability to perform time-aligned speaker change detection.

## 5.1 Multi-Task Learning

We explored various training data configurations for multi-task learning (see Table 2). Row-0 in Table 2 represents how a conventional ST system (i.e., trained on only single-turn ST data) performs under the challenging multi-turn multi-speaker scenario. Other systems in Table 2 yield insights into how to boost the performances by augmenting the training data with auxiliary tasks.

**Joint training of single-turn and multi-turn tasks is beneficial.** Adding multi-turn ST data for training gives marginal improvements (Row-1 vs. Row-0); this suggests that simply adding limited multi-turn data will not suffice for the MT-MS cases. When either single-turn or multi-turn

---

[4]Version: cv-corpus-13.0-2023-03-09.
[5]Signature: nrefs:N|case:lc|eff:no|tok:13a|smooth: exp|version:2.3.1. (Fisher N=4 and CALLHOME N=1).

[6]$max\_grad\_norm = 5.0$.
[7]https://github.com/speechbrain/speechbrain/tree/develop/recipes/LibriSpeech/ASR/transformer

| Training data configuration | | | | Fisher | | CALLHOME | |
|---|---|---|---|---|---|---|---|
| Single-Turn ASR | ST | Multi-Turn ASR | ST | WER (↓) | BLEU (↑) | WER (↓) | BLEU (↑) |
| 0) | ✓ | | | - | 28.3 | - | 8.5 |
| 1) | ✓ | | ✓ | - | 30.9 | - | 8.7 |
| 2) ✓ | ✓ | | | 40.2 | 29.3 | 57.9 | 8.9 |
| 3) | | ✓ | ✓ | 29.4 | 41.5 | 49.9 | 14.7 |
| 4) ✓ | ✓ | ✓ | ✓ | **25.8** | **46.8** | **42.1** | **17.9** |
| 5) ✓ | ✓ | ✓ | | **25.8** | 35.6 | 42.3 | 11.7 |
| 6) ✓ | ✓ | | ✓ | 44.9 | 43.7 | 68.2 | 15.5 |

Table 2: ASR and ST performance of STAC-ST with different training data configurations. Joint training with single-turn and multi-turn data of both ASR and ST tasks achieves the best scores.

| Task tokens | Fisher | | CALLHOME | |
|---|---|---|---|---|
| | WER↓ | BLEU↑ | WER↓ | BLEU↑ |
| [SL], [TL] | 26.4 | 45.0 | 43.7 | 16.6 |
| + [TURN] | **25.8** | 45.2 | 43.1 | 17.6 |
| + [XT] | **25.8** | **46.8** | **42.1** | **17.9** |

Table 3: ASR and ST performance of STAC-ST with the incremental addition of task tokens. Modeling speaker-turn and cross-talk detection with [TURN] and [XT] tokens enhances ASR and MT accuracy.

data has reasonable size (i.e., augmenting ASR data), combining them yields more pronounced improvements (Row-4 vs. Row-2/Row-3). Although single-turn and multi-turn data share the same utterances, split/concatenation-based data augmentation is known to be effective in the low-resource training regime (Nguyen et al., 2021; Lupo et al., 2022).

**Joint training of ST and ASR is beneficial.** Interestingly, training a model with only multi-turn ST data failed to converge, but adding multi-turn ASR data stabilizes the training (Row-3).[8] Moreover, by adding both single-turn and multi-turn ASR data for joint training on top of Row-1, both BLEU and WER are improved by a significant margin (Row-4).

**Multi-turn ASR data helps multi-turn ST.** In our training data, there are more labeled single-turn ST data and multi-turn ASR data than multi-turn ST data. We tested a zero-shot setting where, for the multi-turn condition is only covered by ASR training data (Row-5). Comparing to training with single-turn ST+ASR data only (Row-2), the resulting model brings 3-8 BLEU gains. We hypothesize that, as the encoder is target-language-agnostic, the acoustic representations and the turn detection capacity learned from multi-turn ASR data does partially transfer to the ST task.

**Multi-turn ST does not seem to help multi-turn ASR.** This can be seen by comparing WER scores in Row-2 and Row-6. We hypothesize that the non-monotonicity of the multi-turn ST task disrupts multi-turn ASR performance (Yan et al.,

---

[8]Combining single-turn utterances to create longer (max 30s) multi-turn segments greatly reduces the number of training samples.

2023). However, this can be fixed by adding back multi-turn ASR data (Row-4). Note that we use the Row-4 data configuration for the rest of the paper.

## 5.2 Speaker-Turn and Cross-Talk Detection

The STAC-ST multi-task learning framework also encodes speaker-turn and cross-talk information with task tokens [TURN] and [XT]. We run experiments to study how these task labels impact on ASR and ST performance in MT-MS setting and how they even enable speaker change detection.

**Modeling speaker-turn and cross-talk detection helps multi-speaker ST and ASR.** We run experiments by ablating the two task tokens. Evaluation results in Table 3 show that incrementally adding speaker-turn and cross-talk detection tasks improves translation and transcription quality measured by BLEU and WER. These results support the hypothesis that explicitly learning the two tasks helps the model to better handle MT-MS scenarios.

**Modeling speaker-turn and cross-talk detection enables the model to perform speaker change detection.** The CTC loss helps the encoder to align input audio to text tokens per acoustic frame, including the two task tokens. We trace speaker-turns and cross-talks in the timeline by (1) first running a forward pass on the encoder to extract audio-text temporal alignments and then we (2) locate the spikes of the linear layer on top of the encoder (aka. CTC spikes) only for [TURN] and [XT] tokens. As illustrated in Figure 4, the CTC spikes align remarkably well with actual edges of speaker activities.

By leveraging available annotations in Fisher-CALLHOME test sets, we measure speaker change detection performance with three standard metrics: False Alarm Rate (FAR), Miss Detection Rate (MDR) and F1-score. The FAR computes the rate at which STAC-ST outputs a [TURN] CTC spike

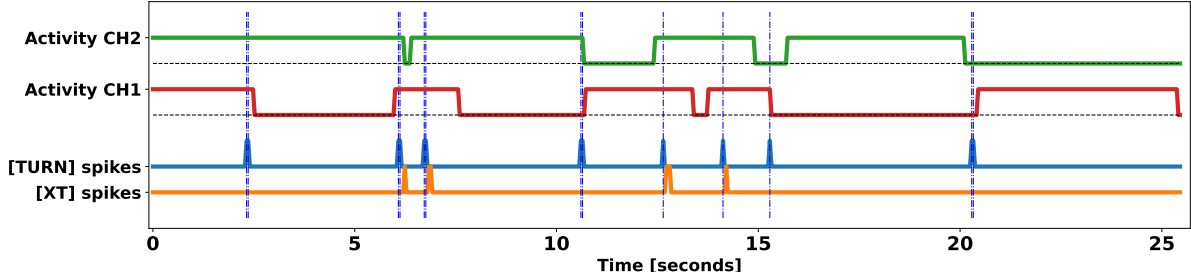

Figure 4: Speaker activity on a Fisher corpus sample. On the top, ground truth human annotation on two audio channels. On the bottom, CTC spikes of turn and cross-talk tokens detected by STAC-ST in the merged channel.

| System | Fisher | | | CALLHOME | | |
|---|---|---|---|---|---|---|
| | F1↑ | MDR↓ | FAR↓ | F1↑ | MDR↓ | FAR↓ |
| PyAnnote | 75.8 | **26.8** | 21.4 | 81.2 | **20.9** | 15.0 |
| STAC-ST | 74.9 | 31.3 | 17.7 | 80.6 | 25.6 | **12.1** |
| STAC-ST (L) | **77.6** | 28.6 | **15.0** | **81.3** | 23.5 | 13.2 |

Table 4: Speaker change detection performance measured by F1, MDR and FAR. We compare STAC-ST with PyAnnote. The strongest L-size STAC-ST model (from Table 5) shows on-par F1-score with PyAnnote. Tolerance is set to 0.25s.

when there are actually no speaker changes. The MDR computed the rate that STAC-ST misses generating [TURN] tokens at speaker changes. While the former two are widely used in speaker segmentation research (Bredin et al., 2020), the F1-score provides an overall assessment of the performance.

To compute these metrics, we first prepare Rich Transcription Time Marked (RTTM) files for each test set from the time-aligned CTC [TURN] spikes. We compared performance of two STAC-ST models (S and L) against a reference system, the speaker segmentation pipeline of the popular PyAnnote toolkit (Bredin and Laurent, 2021).[9] From results listed in Table 4, STAC-ST gets on-par F1-score vs. the reference system in the Fisher-CALLHOME test sets. Using a stronger STAC-ST (L) model improves by 2.5 absolute the F1 score. These results corroborate the importance of the [TURN] task tokens for improving ASR and ST quality.

### 5.3 Benchmarking STAC-ST

We run extensive benchmarks to compare STAC-ST with related work in various settings, including (1) different audio segmentation strategies, (2) model size, and (3) evaluation on single-turn ST.

#### 5.3.1 MT-MS vs. VAD Segmentation

A common practice for translating long-form audio files is to first segment them into smaller chunks based on voice activity detection (VAD). We compare our MT-MS segmentation approach with two popular VAD-based audio segmenters, i.e., WebRTC (Blum et al., 2021) and SHAS (Tsiamas et al., 2022), on the channel-merged Fisher-CALLHOME test sets.[10]

When the audio and reference translation segments are not aligned, like in the case of VAD-based segmentation, the standard process is to first concatenate translation hypotheses and then align and re-segment the conversation-level translation based on the segmented reference translation.[11] However, our preliminary results show that this process yields poor BLEU scores, partially because VAD treats noise as speech, which leads to noisy translation and misalignment. Therefore, we calculate BLEU scores on concatenated hypotheses and references for the whole conversation. BLEU scores in this section are not comparable with the ones reported elsewhere.

As shown in Figure 5, for both Fisher and CALLHOME test sets, BLEU scores of using VAD-based tools (either WebRTC or SHAS) for test data segmentation are below the ones using our MT-MS segmentation. Despite being popular in conventional speech translation, segmenting long-form audio with VAD-based tools is not the best choice for handling multi-talks conversations with speaker-turns. Thus, we resort to using MT-MS segmentation based on human annotations for preparing the test data. This highlights a potential future work direction of producing robust segmentation on noisy long-form conversational audio.

---

[9]https://huggingface.co/pyannote/speaker-segmentation

[10]More details in Appendix F.

[11]mwerSegmenter (Matusov et al., 2005) has been used in IWSLT (Anastasopoulos et al., 2022, 2021) for this purpose.

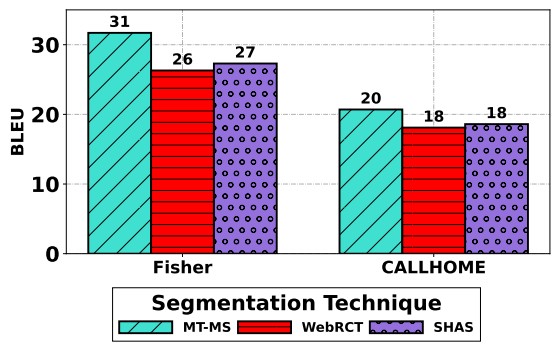

Figure 5: ST performance on Fisher-CALLHOME test data using different segmentation techniques for long-form audio: MT-MS (ours), WebRTC, and SHAS. BLEU scores of using VAD-based tools (either WebRTC or SHAS) for test data segmentation are lower than BLEU computed using our MT-MS segmentation.

| | Fisher | | CALLHOME | |
|---|---|---|---|---|
| Model | WER↓ | BLEU↑ | WER↓ | BLEU↑ |
| Whisper-tiny (39M) | 45.0 | 11.5 | 59.8 | 2.4 |
| Whisper-base (74M) | 36.7 | 29.0 | 49.2 | 8.4 |
| Whisper-small (244M) | 29.1 | 46.7 | **37.9** | 19.2 |
| STAC-ST S (21M) | 25.8 | 46.8 | 42.1 | 17.9 |
| STAC-ST M (86M) | 23.8 | 49.4 | 38.3 | 20.4 |
| STAC-ST L (298M) | **23.5** | **50.0** | 38.5 | **21.0** |

Table 5: ASR and ST performance with increasing model size of STAC-ST and Whisper. STAC-ST achieves better BLEU and WER scores than Whisper with comparable model sizes.

### 5.3.2 Scaled STAC-ST vs. Whisper

Given the lack of prior work on MT-MS ST, we compare STAC-ST against a strong multi-task model, i.e., Whisper (Radford et al., 2023). Whisper is trained with over 2,000 times more speech data than our model (although Fisher-CALLHOME is not included among them) and its smallest version is larger than STAC-ST S. To enable a more fair comparison, we added more speech training data (cf. §4.2) to STAC-ST with size M and L.

Results in Table 5 demonstrate that when we add out-of-domain training data and scale the model accordingly (Kaplan et al., 2020; Bapna et al., 2022; Zhai et al., 2022), STAC-ST achieves better BLEU and WER scores than Whisper with comparable model sizes, although our training data is still three orders of magnitude smaller.

---

12https://github.com/espnet/espnet/tree/master/egs2/fisher_callhome_spanish

| | Fisher | | CALLHOME | |
|---|---|---|---|---|
| Model | WER↓ | BLEU↑ | WER↓ | BLEU↑ |
| Casc. ST (Post et al., 2013) | 36.5 | - | 65.3 | 11.6 |
| Multi-task (Weiss et al., 2017) | 23.2 | 48.7 | 45.3 | 17.4 |
| E2E-ST (Inaguma et al., 2019) | 22.9 | 46.3 | 44.5 | 17.2 |
| ESPnet example (2022)12 | **18.7** | 50.5 | 37.6 | 21.7 |
| Whisper-tiny (39M) | 44.1 | 9.0 | 58.5 | 2.2 |
| Whisper-base (74M) | 34.8 | 25.4 | 48.7 | 6.5 |
| Whisper-small (244M) | 28.1 | 45.3 | 36.5 | 16.8 |
| STAC-ST S (21M) | 20.9 | 49.1 | 36.3 | 20.1 |
| STAC-ST M (86M) | **18.9** | 52.3 | 31.4 | 22.1 |
| STAC-ST L (298M) | **18.8** | **52.6** | **31.0** | **22.4** |

Table 6: ASR and ST performance with the official single-speaker manual segmentation. Previous work results and Whisper baselines are provided. Our strongest model, STAC-ST L yields the best scores.

### 5.3.3 STAC-ST for Single-Turn ST

To position STAC-ST against previous work on ST, we also run experiments under the conventional single-turn ST condition. These experiments enable us to (1) see how our end-to-end multi-task learning approach performs on a specific input condition, and (2) compare STAC-ST against four previous models trained and evaluated on the same task. To allow for comparing results across single-turn and MS-MT conditions, we also report performance with three Whisper systems. Results of these experiments are reported in Table 6. We observe that all our STAC-ST models are competitive with the previous models, also optimized on the Fisher-CALLHOME task. Comparison against the Whisper models confirms the trends observed in Table 5 under the MS-MT condition. Overall, STAC-ST L yields the best BLEU scores on both Fisher and CALLHOME.

## 6 Conclusions

In this work, we present STAC-ST, an end-to-end system designed for single-channel multi-turn & multi-speaker speech translation that uses a multi-task training framework to leverage both ASR and ST datasets. We demonstrate that STAC-ST generalizes to both standard pre-segmented ST benchmarks and multi-turn conversational ST, the latter being a more challenging scenario. STAC-ST is also shown to learn the task of speaker change detection, which helps multi-speaker ST and ASR. We investigate different aspects of STAC-ST, including the impact of model and data size, automatic segmentation for long-form conversational ST, zero-shot multi-turn & multi-speaker ST with-

out specific training data. Overall, this work sheds light on future work towards more robust conversational ST systems that can handle speaker-turns and cross-talks.

## Limitations

1. Our primary test sets, Fisher and CALL-HOME, have narrowly one translation direction (Spanish→English). The only other public conversational ST dataset we are aware of is MSLT (Federmann and Lewis, 2016), but it only contains independent utterances, which is far from representing a realistic MT-MS use case. We call for more publicly available long-form conversational ST data under a friendly license.

2. Due to the same limitation of publicly available datasets, we do only explore conversations between **two** speakers.

3. We segment the test sets based on human annotations. Despite being the best choice for the MT-MS data in our study (§5.3.1), it is not a realistic scenario for testing. We leave improving segmentation on noisy long-form conversational audio as future work.

4. We segment long-form audio files into up to 30s pieces following Radford et al. (2023), but we do not use the preceding segments as context. We focus on improving translation quality of conversations by speaker-turn and cross-talk detection, yet using the context information could also help. In addition, within each MT-MS segment, the inter-utterance context could have already been leveraged (Zhang et al., 2021). We leave analysis of the inter- and intra-segment context as future work.

5. We only test the Transformer architecture as we focus on solving a challenging MT-MS ST task with multi-task learning, which is orthogonal to the architecture choice. We leave exploring other architecture options, such as Conformer (Radfar et al., 2023), HyperConformer (Mai et al., 2023) or Conmer (Radfar et al., 2023) as future work.

## Ethical Considerations

All speech datasets we use have anonymous speakers. We do not have any access to nor try to create any PII (Personal Identifiable Information) of speakers, and our model neither identifies speakers nor uses speaker embeddings.

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

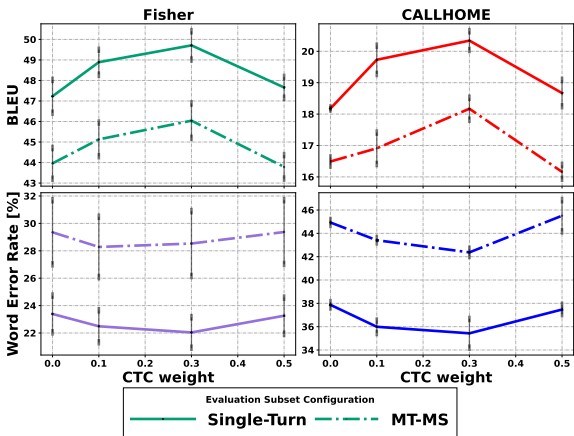

Figure 6: Ablation of the CTC weight in the overall loss computation and its impact in BLEU and WERs for Fisher and CALLHOME development & evaluation sets. Error bars show the standard deviation between dev/dev2/test sets for Fisher and devset/evlset for CALL-HOME. Single-turn and MS-MS results are shown with straight and dashed lines, respectively.

## A    Evaluating Different CTC Weights

In this section, we evaluate different CTC weights for joint ASR & ST training under the `STAC-ST` framework. We show in Figure 6 the results for different S-size models trained on the Fisher-CALLHOME corpora. We confirm that BLEU and WER scores achieve the best with a $\lambda = 0.3$, akin to previous work (Zhang et al., 2022).

## B    Complete Main Evaluation Results on Fisher-CALLHOME

We list complete main results on Fisher-CALLHOME corpora for all the official subsets.

**Multi-Turn Segments.** Table 9 lists BLEU scores for all subsets of Fisher-CALLHOME, while Table 10 lists WER scores.

**Single-Turn Segments.** For the sake of completeness, we also report the performance of `STAC-ST` on each subset of Fisher-CALLHOME with the default utterance segmentation (single-turn). Table 11 lists the BLEU scores, while Table 12 list WER scores.

## C    Impact of Speech Overlap Ratio

In MT-MS data, each segment contains different degree of overlaps. We calculate the overlap ratio for each segment in Fisher and CALLHOME, group the segment-level overlap ratios into 4 bins, and report BLEU scores for each bin in Table 7. The

| Overlap Ratio | Fisher | | CALLHOME | |
|---|---|---|---|---|
| | BLEU | #words | BLEU | #words |
| $x \leq 6\%$ | 48.15 | 10,584 | 21.31 | 4,228 |
| $6\% < x \leq 11\%$ | 47.43 | 7,502 | 19.77 | 3,962 |
| $11\% < x \leq 17\%$ | 45.79 | 6,901 | 16.27 | 4,709 |
| $17\% < x$ | 44.75 | 10,119 | 15.82 | 4,659 |
| all | 46.83 | 39,095 | 17.92 | 18,458 |

Table 7: We calculate the overlap ratio for each segment in Fisher and CALLHOME and then group the segment-level overlap ratios into 4 bins. We report BLEU score and the number of words in reference within each bin.

| TOL | Fisher | | | CALLHOME | | |
|---|---|---|---|---|---|---|
| (s) | F1 | MDR | FAR | F1 | MDR | FAR |
| 0.1 | 58.3 | 46.2 | 36.4 | 67.6 | 37.5 | 26.4 |
| 0.25 | 74.9 | 31.3 | 17.7 | 80.6 | 25.6 | 12.1 |
| 0.5 | 83.4 | 23.0 | 9.0 | 85.5 | 20.8 | 7.2 |
| 1 | 87.3 | 18.4 | 6.2 | 89.3 | 16.2 | 4.5 |

Table 8: Performance of `STAC-ST` on speaker change detection on the multi-turn dataset for all official Fisher-CALLHOME test sets. Tolerance is ablated from 0.1 up to 1 second.

chosen bins are based on [0%, 25%, 50%, 75%, 100%] percentiles found on Fisher and remain the same for CALLHOME. These results correspond to Row-4 in Table 2. We can see that the BLEU score decreases with increasing speech overlaps.

## D    More Examples and Analysis on Speaker-Turn and Cross-Talk Detection

In Figure 7, we provide 3 additional examples of ground-truth speaker activities vs. CTC spikes of `[TURN]` and `[XT]` task tokens (see § 5.2). The title contains the sample ID, transcript and translation together with the `[TURN]` and `[XT]` task tokens.

In Table 8 we evaluate different tolerance values when computing the speaker change detection metrics con both Fisher-CALLHOME test sets. The tolerance (in seconds) allows us to reduce the granularity that we expect in speaker change detection. Giving the fact that `STAC-ST` is not directly optimized for this task, we note that a value of at least 0.25 is critical to reach acceptable scores – by increasing the tolerance from 0.1 to 0.25 seconds, we see a 22% relative increase in F1 score. Setting it to 0.5 seconds further brings a 10% relative improvement.

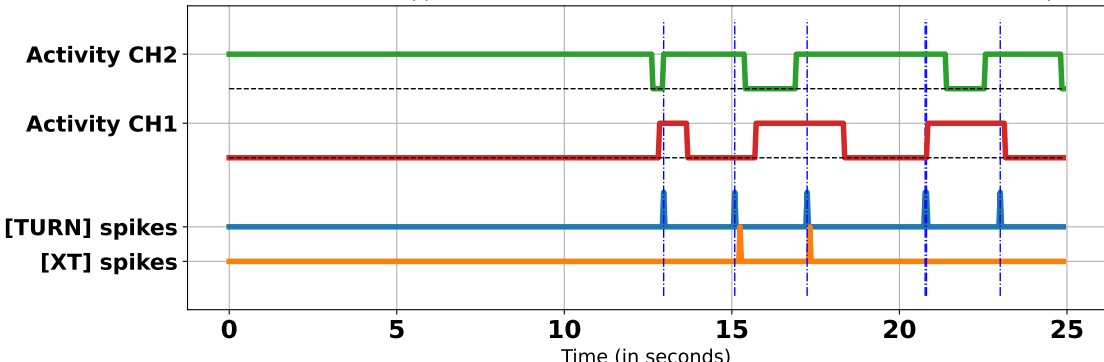

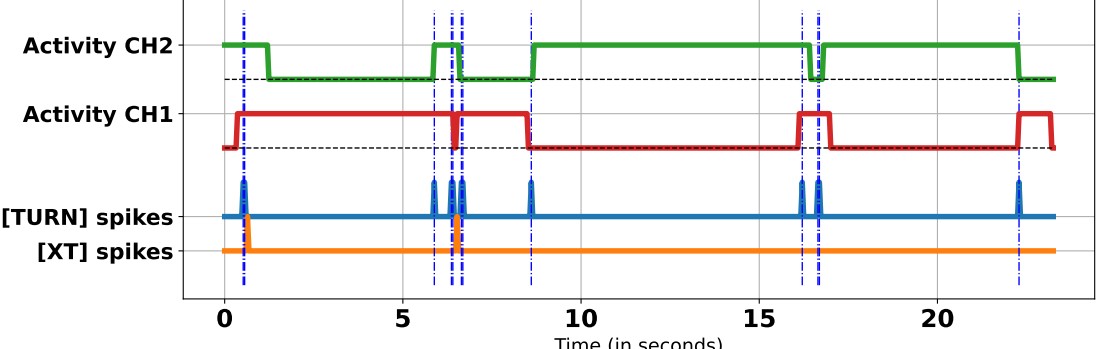

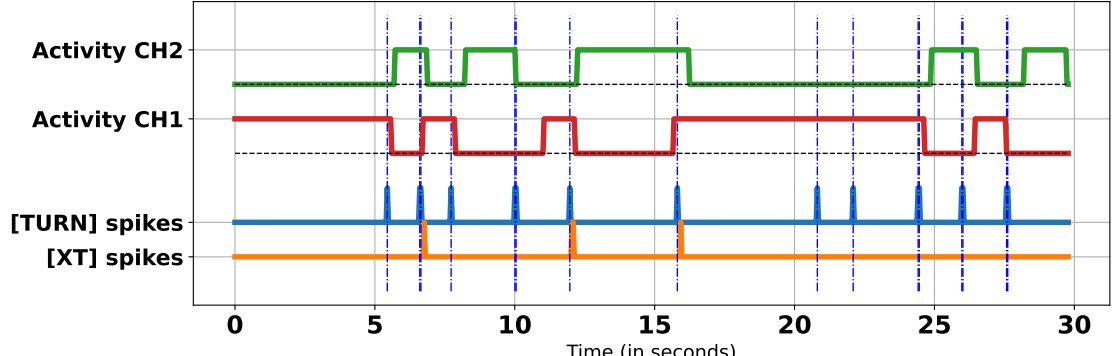

Figure 7: Ground-truth speaker activities and CTC spikes of [TURN] and [XT] task tokens on three randomly selected Fisher samples. The Tile list the ID (recording, file number, start and end time), the ground truth transcript and translation.

| Training Data | | | | BLEU score (↑) | | | | | | |
|---|---|---|---|---|---|---|---|---|---|---|
| Single-turn | | Multi-turn | | Fisher | | | | CALLHOME | | |
| ASR | ST | ASR | ST | dev | dev2 | test | AVG | devtest | evltest | AVG |
|  | ✓ |  |  | 26.2 | 27.0 | 28.3 | 27.2 | 8.6 | 8.5 | 8.5 |
|  | ✓ |  | ✓ | 30.31 | 30.5 | 30.9 | 30.5 | 9.5 | 8.7 | 9.1 |
| ✓ | ✓ |  |  | 25.6 | 27.0 | 29.3 | 27.3 | 8.8 | 8.9 | 8.8 |
|  |  | ✓ | ✓ | 40.2 | 40.0 | 41.5 | 40.5 | 15.0 | 14.7 | 14.8 |
| ✓ | ✓ | ✓ |  | 32.7 | 32.9 | 35.6 | 33.7 | 10.6 | 11.7 | 11.1 |
| ✓ | ✓ |  | ✓ | 42.3 | 42.5 | 43.7 | 42.8 | 15.2 | 15.5 | 15.4 |
| ✓ | ✓ | ✓ | ✓ | 45.1 | 46.1 | 46.8 | 46.0 | 18.4 | 17.9 | 18.2 |

Table 9: BLEU scores on each multi-turn dataset for all the official Fisher-CALLHOME development and test subset. AVG lists the average between dev and test sets.

| Training Data | | | | Word Error Rate (↓) | | | | | | |
|---|---|---|---|---|---|---|---|---|---|---|
| Single-turn | | Multi-turn | | Fisher | | | | CALLHOME | | |
| ASR | ST | ASR | ST | dev | dev2 | test | AVG | devtest | evltest | AVG |
| ✓ |  | ✓ |  | 29.7 | 30.0 | 26.1 | 28.6 | 44.0 | 43.5 | 43.8 |
| ✓ | ✓ |  |  | 45.9 | 46.6 | 40.2 | 44.2 | 58.0 | 57.9 | 58.0 |
|  |  | ✓ | ✓ | 35.2 | 35.8 | 29.4 | 33.5 | 51.4 | 49.9 | 50.7 |
| ✓ | ✓ | ✓ |  | 29.4 | 30.0 | 25.8 | 28.4 | 42.9 | 42.3 | 42.6 |
| ✓ | ✓ |  | ✓ | 52.8 | 54.6 | 44.9 | 50.8 | 64.3 | 68.2 | 66.3 |
| ✓ | ✓ | ✓ | ✓ | 30.2 | 29.6 | 25.8 | 28.5 | 42.6 | 42.1 | 42.4 |

Table 10: WERs on each multi-turn dataset for all the official Fisher-CALLHOME development and test subset. AVG lists the average between dev and test sets.

| Training Data | | | | BLEU score (↑) | | | | | | |
|---|---|---|---|---|---|---|---|---|---|---|
| Single-turn | | Multi-turn | | Fisher | | | | CALLHOME | | |
| ASR | ST | ASR | ST | dev | dev2 | test | AVG | devtest | evltest | AVG |
|  | ✓ |  |  | 46.7 | 47.3 | 46.5 | 46.8 | 18.7 | 18.9 | 18.8 |
|  | ✓ |  | ✓ | 34.1 | 34.5 | 34.3 | 34.3 | 11.4 | 11.0 | 11.2 |
| ✓ | ✓ |  |  | 50.2 | 51.5 | 50.0 | 50.5 | 21.2 | 21.2 | 21.2 |
|  |  | ✓ | ✓ | 41.1 | 41.6 | 41.7 | 41.4 | 14.8 | 14.9 | 14.8 |
| ✓ | ✓ | ✓ |  | 47.5 | 48.1 | 47.1 | 47.5 | 18.5 | 19.2 | 18.8 |
| ✓ | ✓ |  | ✓ | 47.2 | 47.7 | 46.6 | 47.2 | 19.4 | 18.6 | 19.0 |
| ✓ | ✓ | ✓ | ✓ | 49.6 | 50.4 | 49.1 | 49.7 | 20.5 | 20.1 | 20.3 |

Table 11: BLEU scores on each single-turn dataset for all the official Fisher-CALLHOME development and test subset. AVG lists the average between dev and test sets.

| Training Data | | | | Word Error Rate (↓) | | | | | | |
|---|---|---|---|---|---|---|---|---|---|---|
| Single-turn | | Multi-turn | | Fisher | | | | CALLHOME | | |
| ASR | ST | ASR | ST | dev | dev2 | test | AVG | devtest | evltest | AVG |
| ✓ |  | ✓ |  | 23.5 | 22.8 | 21.0 | 22.5 | 35.5 | 36.3 | 35.9 |
| ✓ | ✓ |  |  | 22.8 | 22.2 | 20.7 | 21.9 | 34.0 | 34.6 | 34.3 |
|  |  | ✓ | ✓ | 31.5 | 31.6 | 27.9 | 30.3 | 48.4 | 48.4 | 48.4 |
| ✓ | ✓ | ✓ |  | 23.1 | 22.5 | 20.8 | 22.1 | 35.2 | 35.6 | 35.4 |
| ✓ | ✓ |  | ✓ | 26.0 | 26.1 | 23.4 | 25.2 | 38.7 | 39.7 | 39.2 |
| ✓ | ✓ | ✓ | ✓ | 23.0 | 22.2 | 20.8 | 22.0 | 34.6 | 36.3 | 35.4 |

Table 12: WERs on each single-turn dataset for all the official Fisher-CALLHOME development and test subset. AVG lists the average between dev and test sets.

| Special Tokens | Fisher | | | | CALLHOME | | |
|---|---|---|---|---|---|---|---|
| | dev | dev2 | test | AVG | devtest | evltest | AVG |
| **BLEU score (↑)** | | | | | | | |
| N/A | 43.4 | 44.2 | 45.0 | 44.2 | 17.0 | 16.6 | 16.8 |
| [TURN] | 44.2 | 44.7 | 45.2 | 44.7 | 17.6 | 17.6 | 17.6 |
| [TURN] + [XT] | **45.1** | **46.1** | **46.8** | **46.0** | **18.4** | **17.9** | **18.1** |
| *Word Error Rate (↓)* | | | | | | | |
| N/A | 29.9 | 30.3 | 26.4 | 28.9 | 43.9 | 43.7 | 43.6 |
| [TURN] | **29.2** | 31.1 | **25.8** | 28.7 | 43.2 | 43.1 | 43.2 |
| [TURN] + [XT] | 30.2 | **29.6** | **25.8** | **28.5** | **42.6** | **42.1** | **42.4** |

Table 13: Ablation of the impact of encoding speaker turn and cross-talk information with [TURN] and [XT]. BLEU scores and WERs are listed for multi-turn dataset for all the official Fisher-CALLHOME development and test sets. AVG lists the average between dev and test sets.

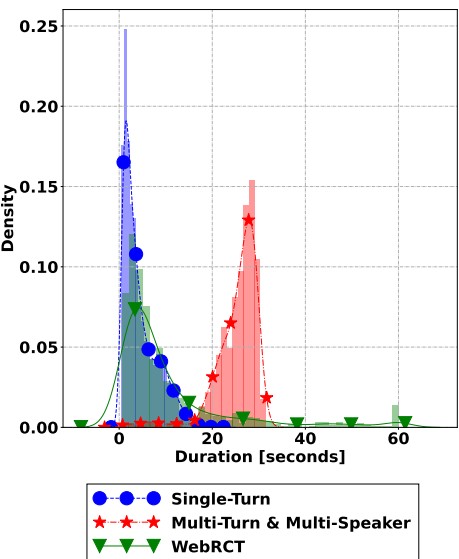

Figure 8: Data distribution for Fisher test set with different segmentation approaches.

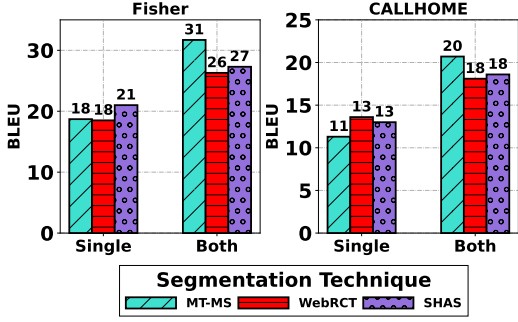

Figure 9: We compare different segmentation techniques with two training data configurations: only **Single**-turn data and **Both** single-turn and multi-turn data. The bars denote different segmentation techniques for long-form audio, including MT-MS segmentation (proposed in this work), VAD via WebRTC (Blum et al., 2021) or SHAS (Tsiamas et al., 2022).

# E   Complete Ablation Results for [TURN] & [XT] Task Tokens

We provide compete ablation results of adding [TURN] & [XT] task tokens on all the official development and test sets of Fisher-CALLHOME, as listed in Table 13.

# F   More Details of VAD-Based Segmentation

With WebRTC, audio is split when 90% of consecutive frames do not include speech. We set the frame length to 30 ms and the aggressiveness parameter to 1 as in (Tsiamas et al., 2022). With SHAS, we set 1-30 as the min-max sequence length.

SHAS was trained on monologue corpora with MuST-C (Di Gangi et al., 2019). Thus, we per-

form an additional pre-processing step to minimize the domain mismatch between SHAS and Fisher-CALLHOME. (1) We extract the speech activity boundaries for each audio file from the original metadata. (2) We modify each audio file by masking with 0 all the regions in the signal where there is no speech activity, i.e., setting all the non-speech activity regions to silence. (3) We then use the masked long-form audio files with SHAS. This step decreases the false alarms rate that can be produced by SHAS on noisy segments or between contiguous utterances where there are close-talks. Close-talks are areas where two utterances are too close and the segmentation tools might not generalize well. In order to keep comparable the experimental and evaluation setup, we perform the same pre-processing step when using WebRTC.

Besides SHAS (Figure 3), we also plot the segmentation distribution of WebRTC on the Fisher

| System | Translation |
|---|---|
| Reference | ... hello good evening who is this [TURN] [XT] how's it going hey this is guillermo ... |
| Baseline | ... hello good evening how are you i'm guillermo ... |
| STAC-ST | ... hello good evening who is it [TURN] [XT] how is it going eh i'm guillermo ... |

Table 14: In this example, the second speaker jumps in while the first speaker is saying "who is this". The baseline model trained with only single-turn data fails to handle the cross-talk and cuts off "who is this". Our STAC-ST model not only accurately identifies the speaker-turn change and cross-talk (by producing [TURN] [XT]), but also successfully serializes the cross-talk.

| Model | Size ($\theta$) | Fisher | | | | CALLHOME | | |
|---|---|---|---|---|---|---|---|---|
| | | dev | dev2 | test | AVG | devtest | evltest | AVG |
| **BLEU score ($\uparrow$)** | | | | | | | | |
| Whisper-tiny | 39M | 8.1 | 7.5 | 11.5 | 9.0 | 1.9 | 2.4 | 2.2 |
| Whisper-base | 74M | 27.4 | 23.7 | 29.0 | 26.7 | 7.3 | 8.4 | 7.9 |
| Whisper-small | 244M | 44.2 | 44.1 | 46.7 | 45.0 | 19.2 | 19.2 | 19.2 |
| Whisper-medium | 769M | 48.6 | 47.7 | 49.2 | 48.5 | 22.5 | 23.1 | 22.8 |
| STAC-ST (S) | 21M | 45.1 | 46.1 | 46.8 | 46.0 | 18.4 | 17.9 | 18.2 |
| STAC-ST (M) | 86M | 48.1 | 48 | 49.4 | 48.5 | 20.2 | 20.4 | 20.3 |
| STAC-ST (L) | 298M | 48.6 | 48.9 | 50.0 | 49.2 | 21.0 | 21.0 | 21.0 |
| **Word Error Rate ($\downarrow$)** | | | | | | | | |
| Whisper-tiny | 39M | 51.5 | 50.1 | 45.0 | 48.9 | 60.3 | 59.8 | 60.1 |
| Whisper-base | 74M | 41.8 | 42.0 | 36.7 | 40.2 | 50.0 | 49.2 | 49.6 |
| Whisper-small | 244M | 33.9 | 33.7 | 29.1 | 32.2 | 39.1 | 37.9 | 38.5 |
| Whisper-medium | 769M | 31.3 | 30.9 | 28.7 | 30.3 | 33.9 | 32.3 | 33.1 |
| STAC-ST (S) | 21M | 30.2 | 29.6 | 25.8 | 28.5 | 42.6 | 42.1 | 42.4 |
| STAC-ST (M) | 86M | 27.0 | 28.1 | 23.8 | 26.3 | 40.1 | 38.3 | 39.2 |
| STAC-ST (L) | 298M | 27.9 | 27.9 | 23.5 | 26.4 | 38.98 | 38.5 | 38.7 |

Table 15: Comparison between Whisper vs scaled STAC-ST using more training data. WER and BLEU scores are reported on the multi-turn dataset for all the official Fisher-CALLHOME development and test subsets. AVG lists the average between dev and test sets.

test set in Figure 8. WebRTC yields a more reasonable distribution than SHAS. Note that some samples are longer than 30 seconds.

We compare different segmentation techniques with two training data configurations in Figure 9: only **Single**-turn data, i.e., Row-2 in Table 2; **Both** single-turn and multi-turn data, i.e., Row-4 in Table 2. Using our proposed configuration, Both, helps all segmentation techniques we tested during inference.

## G  Example Translations With and Without using STAC-ST

We provide example translations with and without using STAC-ST in Table 14.

## H  Complete Results of Scaled STAC-ST vs. Whisper

We list complete evaluation results of scaled STAC-ST vs. Whisper for the MT-MS Fisher-CALLHOME development and test sets in Table 15.

## I  Complete Results of STAC-ST for Single-Turn ST

We list complete evaluation results of STAC-ST vs. prior work for the single-turn Fisher-CALLHOME development and test sets in Table 16. Note that in the main paper, i.e., Table 6, we only list (1) the work that released the Fisher-CALLHOME corpora (i.e., Casc. ST) and (2) the top three models that report both WER and BLEU scores (i.e., Multi-task, E2E-ST, ESPnet example).

| Model | Size ($\theta$) | Fisher | | | | CALLHOME | | |
|---|---|---|---|---|---|---|---|---|
| | | dev | dev2 | test | AVG | devtest | evltest | AVG |
| **BLEU score ($\uparrow$)** | | | | | | | | |
| Cas. ASR-MT (Post et al., 2013) | - | - | 35.5 | - | - | - | 11.6 | - |
| Multi-task ASR/ST (Weiss et al., 2017) | | 48.3 | 49.1 | 48.7 | 48.7 | 16.8 | 17.4 | 17.1 |
| E2E-ST M2Mc† (Inaguma et al., 2019) | | 44.1 | 45.4 | 45.2 | 44.9 | 16.4 | 16.2 | 16.3 |
| EMc2+ASR-PT† (Inaguma et al., 2019) | | 46.3 | 47.1 | 46.3 | 46.6 | 17.3 | 17.2 | 17.3 |
| E2E-ST streaming (Deng et al., 2022) | | 47.9 | 48.2 | 47.7 | 47.9 | 15.5 | 15.3 | 15.4 |
| ESPnet example (2022) | | 51.8 | 52.3 | 50.5 | 51.5 | 22.3 | 21.7 | 22.0 |
| Whisper-tiny | 39M | 7.4 | 5.6 | 9.0 | 7.3 | 2.0 | 2.2 | 2.1 |
| Whisper-base | 74M | 19.1 | 20.4 | 25.4 | 21.6 | 6.0 | 6.5 | 6.2 |
| Whisper-small | 244M | 45.4 | 40.7 | 45.3 | 43.8 | 17.5 | 16.8 | 17.1 |
| Whisper-medium | 769M | 51.7 | 49.2 | 48.8 | 49.9 | 23.5 | 23.5 | 23.5 |
| STAC-ST (S) | 21M | 49.6 | 50.4 | 49.1 | 49.7 | 20.5 | 20.1 | 20.3 |
| STAC-ST (M) | 86M | 52.0 | 51.9 | 52.3 | 52.1 | 23.0 | 22.1 | 22.6 |
| STAC-ST (L) | 298M | 52.4 | 52.8 | 52.6 | 52.6 | 22.7 | 22.4 | 22.5 |
| **Word Error Rate ($\downarrow$)** | | | | | | | | |
| SAT-fMLLR (Post et al., 2013) | | 41.3 | 40.0 | 36.5 | 39.3 | 64.7 | 65.3 | 65.0 |
| SAT-SGMM (Kumar et al., 2014b) | | 35.9 | 34.5 | - | - | - | - | - |
| Multi-task ASR/ST (Weiss et al., 2017) | | 25.7 | 25.1 | 23.2 | 24.7 | 44.5 | 45.3 | 44.9 |
| E2E-ST M2Ma† (Inaguma et al., 2019) | | 25.6 | 25.0 | 22.9 | 24.5 | 43.5 | 44.5 | 44.0 |
| Joint ASR+MT (Soky et al., 2022) | | 22.8 | 22.3 | 20.5 | 21.9 | 39.5 | 39.4 | 39.5 |
| ESPnet example (2022) | | 20.5 | 20.2 | 18.7 | 19.8 | 37.8 | 37.6 | 37.7 |
| Whisper-tiny | 39M | 50.9 | 49.9 | 44.1 | 48.3 | 60.5 | 58.5 | 59.5 |
| Whisper-base | 74M | 41.4 | 39.5 | 34.8 | 38.6 | 49.0 | 48.7 | 48.8 |
| Whisper-small | 244M | 32.2 | 30.5 | 28.1 | 30.2 | 36.9 | 36.5 | 36.7 |
| Whisper-medium | 769M | 28.3 | 26.8 | 25.8 | 27.0 | 29.8 | 29.3 | 29.6 |
| STAC-ST (S) | 21M | 23.0 | 22.2 | 20.9 | 22.0 | 34.6 | 36.3 | 35.4 |
| STAC-ST (M) | 86M | 21.1 | 20.4 | 18.9 | 20.1 | 30.2 | 31.4 | 30.8 |
| STAC-ST (L) | 298M | 21.0 | 20.6 | 18.8 | 20.1 | 30.4 | 31.0 | 30.7 |

Table 16: Comparison between previous work vs. scaled STAC-ST. WER and BLEU scores are reported on single-turn segments of all the official Fisher-CALLHOME development and test subsets. AVG lists the average between dev and test sets. We list the best BLEU/WER scores for each model from previous work. In some cases, it includes ASR or MT pre-training. †Multilingual model, name convention in (Inaguma et al., 2019).

## J  Additional Results on CoVoST 2

Traditional speech translation datasets are composed of single-turn pre-segmented utterances. Following Section 5.3.3, we also run experiments on the CoVoST 2 test set.[13] In the following Table 17, we report BLEU scores on 3 translation directions (German/French/Spanish→English) and compare with 3 recent papers that report BLEU scores on CoVoST 2: Whisper (Radford et al., 2023), XLS-R (Babu et al., 2022), and CoVoST2 (Wang et al., 2021).[14] We list our STAC-ST models ranging from S-size to L-size. They were trained on CoVoST 2 ST and Common Voice ASR data with both single-turn and synthetic multi-turn segmen-

tations as introduced in Section 4.2. The Fisher-CALLHOME training data was also used for the Spanish→English model. Whisper, XLS-R and CoVoST2 A2E-M are multilingual models. For fair comparison, we trained a multilingual STAC-ST L-size model by combining data of all related languages. Our languages tokens specify the translation direction.

The results show that (1) our multilingual large model outperforms Whisper and XLS-R multilingual models with comparable sizes, even though Whisper and XLS-R where trained on data two orders of magnitude larger: 680k hours for Whisper, 436k hours for XLS-R, and 3k hours for STAC-ST L multilingual; (2) our models with smaller sizes sometimes outperform larger Whisper mod-

[13]We used Common Voice version 13.0 to create the data.
[14]CoVoST2 reports case-sensitive BLEU.

| | Multilingual? | Model Size | DE → EN | FR → EN | ES → EN |
|---|---|---|---|---|---|
| *Baselines* | | | | | |
| (1)Whisper-base (Radford et al., 2023) | Y | 74M | 11.7 | 15.4 | 21.3 |
| (2)Whisper-small (Radford et al., 2023) | Y | 244M | 25.3 | 27.3 | 33.0 |
| (3)XLS-R (Babu et al., 2022) | Y | 300M | 26.7 | 32.9 | 34.1 |
| (4)CoVoST2 Bi-ST (Wang et al., 2021) | - | – | 17.1 | 26.3 | 23.0 |
| (5)CoVoST2 A2E-M (Wang et al., 2021) | Y | – | 18.9 | 27.0 | 28.0 |
| *Ours* | | | | | |
| (6) STAC-ST S | - | 21M | 19.7 | 29.2 | 29.1 |
| (7) STAC-ST M | - | 86M | 20.5 | 31.8 | 32.6 |
| (8) STAC-ST L | - | 298M | 21.4 | 25.2 | 33.0 |
| (9) STAC-ST L - Multilingual | Y | 298M | **27.5** | **34.0** | **35.8** |

Table 17: BLEU scores on three language directions of the CoVoST 2 corpus test set (Wang et al., 2021). The results show that (1) our multilingual large model outperforms Whisper and XLS-R multilingual models with comparable sizes, even though Whisper and XLS-R where trained on data two orders of magnitude larger; (2) our models with smaller sizes sometimes outperform larger Whisper models, such as STAC-ST 21M vs. Whisper 244M on French→English.

els, such as STAC-ST 21M vs. Whisper 244M on French→English. These results along with our main paper demonstrate that our proposed approach is well-suited for both the novel single-channel multi-speaker speech translation task and the conventional pre-segmented speech translation.