# OpenReview forum: "End-to-End Single-Channel Speaker-Turn Aware Conversational Speech Translation"
_EMNLP/2023/Conference — EMNLP 2023 Main_

### Official Review · Reviewer_4M4i · 2023-08-01

**Soundness:** 4

**Excitement:**

4: Strong: This paper deepens the understanding of some phenomenon or lowers the barriers to an existing research direction.

**Paper Topic And Main Contributions:**

The paper introduce an end-to-end speech translation system that can translate multi-talker conversations. The model is trained to perform ASR or speech translation of speech segments of up to 30 seconds containing speech from two speakers, with overlap. The authors have explored using task tokens such as source and target languages and turn and overlap prediction to perform multi-tasks with a single model.




**Questions For The Authors:**

Below are some comments
1.	In section 3.2, the description of the auxiliary tasks should be included. Currently, the paper only mentions that task tokens are inserted to specify the task. Still, there is no description of how the turn prediction and cross-talk detection tasks are implemented. I assume that the output simply provides a serialized output containing [Turn] and [cross-talk] output tokens, but this should be explicitly explained, and also how the labels are generated should be mentioned. Probably showing a figure would make this part clearer.
2.	In Table 1, what is speech act. [%]. Is it the speech activity? If so, please explain it in the text. Besides, an essential statistic would be the speech overlap ratio.
3.	In Section 4.1, the authors segment the data in chunks of up to 30 seconds using the true segmentation information. According to the limitation section, it seems that this procedure is performed not only during training but also for testing. This should be clearly mentioned in Section 4.1.
4.	Figure 4, I could not perceive from the figure that the turn and cross-talk tokens were correctly predicted. This is related to comment 1., but without a clear explanation of how the turn and cross-talk task is implemented and what should be the correct output. Maybe you could add the reference, including the [turn] and [cross talk] tokens above the figure.
5.	Table 4: could you estimate your system in terms of diarization performance and provide DER values?
6.	Could you evaluate the performance of different speech overlaps?
7.	In Tables 5 and 6, bold fonts do not always seem to reflect the best performance. Could you double-check?


**Reasons To Accept:**

This work may be one of the first works tackling E2E multi-speaker translation. The techniques' novelty may be limited, but the problem, task definition, and experimental investigations are novel enough for publication. Besides, the authors have carried out extensive experiments showing the potential of their approach.


**Reasons To Reject:**

The paper is relatively clearly written, except that the handling of multi-turn is explained very vaguely and could be refined.


**Reproducibility:**

3: Could reproduce the results with some difficulty. The settings of parameters are underspecified or subjectively determined; the training/evaluation data are not widely available.

**Reviewer Confidence:**

3: Pretty sure, but there's a chance I missed something. Although I have a good feel for this area in general, I did not carefully check the paper's details, e.g., the math, experimental design, or novelty.

**Typos Grammar Style And Presentation Improvements:**

There are a few typos, e.g., “is know.” Could you double-check the manuscript for typos and grammar mistakes?

---

> ### Author Rebuttal · Authors · 2023-08-28
>
> Thanks for your time and insightful reviews.
>
> Please refer to our answers to your specific questions which are assumed to instantiate “the handling of multi-turn is explained very vaguely” (reason to reject).
>
> Question 1 (How the turn prediction and cross-talk detection tasks are implemented? How the labels are generated?):
> - Yes, the auxiliary tasks of detecting speaker-turn changes and cross-talks are implemented by **training** with serialized text containing [TURN] and [XT], and the model learns to predict these two tokens and serialize the output by re-ordering target words in overlaps during **inference**. We will clarify this in Section 3.2.
> - Section 3.2 describes the procedure of generating labels including task tokens. We serialize transcripts or translations and put [TURN] and [XT] between utterances when needed. We will add a figure to illustrate this procedure more clearly.
>
> Question 2 (What is speech act.? An essential statistic would be the speech overlap ratio.):
> - Yes, it is the speech activity. We will explain it in the caption.
> - We will add the following statistics of speech overlap ratio to Table 1.
> - |Fisher Statistics|train| dev|dev2|test|
> |-----------------|-----|----|----|----|
> |Overlap ratio (%)| 12.7|14.5|16.8|11.2|
> - |CALLHOME Statistics|train| dev|test|
> |-------------------|-----|----|----|
> |Overlap ratio (%)  | 11.7|14.6|11.8|
>
> Question 3 (segment the data using the true segmentation for testing should be clearly mentioned in Section 4.1):
> - Thank you for the suggestion. Besides in the limitation section, we mentioned in Section 5.3.1 that “we resort to using MT-MS segmentation based on human annotations for preparing the test data”. We would also clarify it in Section 4.1.
>
> Question 4 (Figure 4, I could not perceive from the figure that the turn and cross-talk tokens were correctly predicted.):
> - As discussed in comment 1, for the speech translation task, the model just needs to correctly serialize the text and put [TURN] and [XT] tokens in the right position.
> - [TURN] and [XT] tokens do not carry exact time information, and the ability of time-aligned speaker change detection shown in Figure 4 was learned implicitly by the model. In fact, we do not provide ground-truth time stamps for training this intermediate task. We’ll add the reference to Figure 4 for comparison.
>
> Question 5 (Could you estimate your system in terms of diarization performance and provide DER values?):
> - The system does not perform diarization, which involves clustering segments according to speaker identity. Our model learns speaker-turn and cross-talk information implicitly, which would not provide speaker-related information for evaluating the diarization performance.
>
> Question 6 (Could you evaluate the performance of different speech overlaps?):
> - This is a great suggestion! We group Fisher and CALLHOME test segments into 4 bins by their average ratio of speech overlaps and report BLEU for each bin.
> - Fisher:
>
> |Overlap ratio|BLEU|
> |------------:|---:|
> |        x<=6%|48.2|
> |    6%<x<=11%|47.4|
> |   11%<x<=17%|45.8|
> |        17%<x|44.8|
> |          all|46.8|
>
> - CALLHOME:
>
> |Overlap ratio|BLEU|
> |------------:|---:|
> |        x<=6%|21.3|
> |    6%<x<=11%|19.8|
> |   11%<x<=17%|16.3|
> |        17%<x|15.8|
> |          all|17.9|
>
> - These results correspond to Row-3 in Table 2. The BLEU score decreases with increasing speech overlaps.
>
> Question 7 (In Tables 5 and 6, bold fonts do not always seem to reflect the best performance.):
> - Thanks for the catch! 37.9 of WER on CALLHOME in Table 5 should be bold.
> - We highlight numbers that are better than the second place by at least 0.2, that is why we bold 3 numbers of WER on Fisher in Table 6.
>
> Typos Grammar Style
> - Thank you for the detailed check. We will check typos and grammar mistakes carefully.

---

### Official Review · Reviewer_JKE7 · 2023-08-04

**Soundness:** 4

**Excitement:**

3: Ambivalent: It has merits (e.g., it reports state-of-the-art results, the idea is nice), but there are key weaknesses (e.g., it describes incremental work), and it can significantly benefit from another round of revision. However, I won't object to accepting it if my co-reviewers champion it.

**Paper Topic And Main Contributions:**

This paper investigates conversational speech translation using an end-to-end framework. The multiple tasks of automatic speech recognition, speech translation and speaker turn detection, are combined in a multi-task training method. The end-to-end model is trained to generate specific tokens in a serialized labeling format. Such a combination is novel, the experimental results also demonstrate the effectiveness of the proposed framework.

**Reasons To Accept:**

The experimental results demonstrate the effectiveness of the proposed framework. The visualization of CTC spikes show that the system can model the speaker activities well.

**Reasons To Reject:**

It is unclear if there is overlapped speech in the conversation, how to predict the tokens in a serialized format.

**Reproducibility:**

4: Could mostly reproduce the results, but there may be some variation because of sample variance or minor variations in their interpretation of the protocol or method.

**Reviewer Confidence:**

4: Quite sure. I tried to check the important points carefully. It's unlikely, though conceivable, that I missed something that should affect my ratings.

---

> ### Author Rebuttal · Authors · 2023-08-28
>
> Thanks for your time and insightful reviews.
>
> Reasons To Reject (It is unclear if there is overlapped speech in the conversation, how to predict the tokens in a serialized format.):
> - As introduced in Section 3.2, “if utterances $u_{t}$ and $u_{t+1}$ overlap in time, we append the targets of utterance $u_{t+1}$ after utterance $u_{t}$. The order of utterances is determined by their start time.” So the model generates the serialized output in a "First In First Out (FIFO)" manner — it first predicts the tokens in the former utterance, then adds [TURN] [XT] and predicts the tokens in the latter utterance. We will add a figure to illustrate the procedure mentioned in Section 3.2 more clearly.
> - Serializing the overlapped transcriptions/translations is a word re-ordering problem, and the model uses the cross-attention mechanism to predict the latter utterance tokens after [TURN] [XT].

---

### Official Review · Reviewer_3SN2 · 2023-08-10

**Typos Grammar Style And Presentation Improvements:** "cross-talks" -> cross-talk
**Soundness:** 3

**Excitement:**

4: Strong: This paper deepens the understanding of some phenomenon or lowers the barriers to an existing research direction.

**Missing References:**

You tend to cite arxiv versions of papers, for which peer-reviewed versions are available. E.g., for Chan et al. 2015 it would be better to cite the ICASSP 2016 version.

**Paper Topic And Main Contributions:**

The paper describes a multitask framework for speech translation that is specifically target towards speech with multiple speakers per channel. To address this condition it pays special attention to speaker turns and cross-talk detection, by using specific tokens in the training data. Training data is simulated by collapsing the separate channels of training data such as Switchboard, into one channel.
A comprehensive set of experiments is conducted to measure the influence of single turn and multi turn data for the ASR and Speech Translation task, showing the benefits of multi-task training and training on single-turn and multi-turn data.

**Questions For The Authors:**

Question A: Is there a particular reason why in Figure 4 [XT] does not spike around 0s?

**Reasons To Accept:**

Interesting set-up for a relevant problem with comprehensive experiments supporting the claims of the paper.

**Reasons To Reject:**

Some decisions could be better motivated. E.g., the authors only us a cross-talk tag, that learns the beginning of cross-talk, but not when cross talk ends. It is not clearly motivated how this benefits speech translation. In the case of longer portions of cross-talk it is not clear how WER and BLEU are evaluated during speaker overlap, as in principle one could have two parallel references during the time of the overlap.

**Reproducibility:**

4: Could mostly reproduce the results, but there may be some variation because of sample variance or minor variations in their interpretation of the protocol or method.

**Reviewer Confidence:**

5: Positive that my evaluation is correct. I read the paper very carefully and I am very familiar with related work.

---

> ### Author Rebuttal · Authors · 2023-08-28
>
> Thanks for your time and insightful reviews.
>
> Reasons To Reject (the authors only use a cross-talk tag, that learns the beginning of cross-talk, but not when cross talk ends):
> - By placing the cross-talk token **after** the speaker-turn token, we enforce the model to differentiate the **beginning** of cross-talk and non-cross-talk utterances, so the model learns to delay outputting the utterance causing cross-talk and serialize the translation. Our quantitative evaluation verified the effectiveness of this serialization strategy.
> - Using a token to specify the **end** of cross-talk is an interesting idea that probably works as well. In many cases, it equals to placing the cross-talk token **before** the speaker-turn token leading a cross-talk (e.g. the example bellow where word2 and WORD2 overlap in speech.) We leave empirical comparison of these two almost equivalent serialization strategies as future work.
> ```
> CHANNEL 1: |word1 word2|
> CHANNEL 2:       |WORD2 WORD3 ...|
>
> Serialization 1: word1 word2 [TURN] [XT-START] WORD2 WORD3 ...
> Serialization 2: word1 word2 [XT-END]   [TURN] WORD2 WORD3 ...
> ```
> - Both system output and reference text are serialized, so there are no overlaps and parallel references in text. Serialized text is evaluated in a conventional way.
>
> Question (Is there a particular reason why in Figure 4 [XT] does not spike around 0s?):
> - We use [TURN] for speaker-turn **changes**, and [XT] always follows [TURN], so [TURN] and [XT] spikes correspond to speaker activity **changes**. We do not prepend [XT] to a segment, so there is no [XT] spike around 0s.
>
> Missing References (You tend to cite arxiv versions of papers.):
> - We will refresh the bibliography to reflect peer-reviewed versions if available.

---

### Official Review · Reviewer_WTTK · 2023-08-11

**Soundness:** 3

**Excitement:**

3: Ambivalent: It has merits (e.g., it reports state-of-the-art results, the idea is nice), but there are key weaknesses (e.g., it describes incremental work), and it can significantly benefit from another round of revision. However, I won't object to accepting it if my co-reviewers champion it.

**Paper Topic And Main Contributions:**

The authors present a turn-aware transformer-based model for conversational speech translation and asr, using a single audio channel.

"TURN" and "XT" tokens are introduced into the model (figure 2) to indicate speaker turns (which are provided during training, and I believe inferred at test time), while "SL" and "TL" tokens are introduced to indicate translation or asr transcription modes (which are fixed and known for each conversation segment that is decoded, during both train and test).

Experimental results on Fisher and CALLHOME show that:
1) joint training with both single and multi-turn data as well as training on both translation and asr data improves performance (table 2),
2) Addition of the "TURN" and "XT" tokens slightly improves ST and WER performance (table 3),
3) Speaker change detection results showing slightly better performance than PyAnnotate (table 4),
4) Better ST performance using their conversation segmentation/chunking technique than WebRCT and SHAS (table 5),
5) Better ASR and ST performance than Whisper models of similar sizes on Fisher and CALLHOME, despite using 3 orders of magnitude less data (table 5), and
6) Strong single speaker performance relative to Whisper and existing techniques (table 6).


**Questions For The Authors:**

See previous section.

**Reasons To Accept:**

- Well written paper. Study seems quite thorough (with some caveats, see limitations section).
- Experiments demonstrate the utility of their model, as described in the summary.

**Reasons To Reject:**

- While the advantage of the TURN and XT tokens is shown in table 3, the merit of the SL and TL tokens is not reported on. If all of the new tokens are omitted, what is the performance of their system on the same training data? This seems like an important baseline.
- Comparing the single speaker and conversational results of table 6 and 5, it seems that Whisper is completely robust to the conversational vs. single speaker scenario, whereas the presented system degrades signficantly. This warrants discussion.
- The single speaker results from other papers are all 4 or more years old. Are these results truly representative of the SOTA? Also, the size of these models is not provided.
- The single speaker results also suggest that the Whisper system may not be well matched to the Fisher and CALLHOME data. Shouldn't finetuning these Whisper models on the training segment of this data (single only and/or conversational) be the SOTA baseline?
- In the paper, the statements around utilizing human annotations for segmenting the test set are a bit confusing. Please confirm that this refers only to using gt for the "chunking" of the test data for processing, as opposed to providing turn segmentations to the model, and please make this more clear in the paper.

**Reproducibility:**

4: Could mostly reproduce the results, but there may be some variation because of sample variance or minor variations in their interpretation of the protocol or method.

**Reviewer Confidence:**

3: Pretty sure, but there's a chance I missed something. Although I have a good feel for this area in general, I did not carefully check the paper's details, e.g., the math, experimental design, or novelty.

**Typos Grammar Style And Presentation Improvements:**

spikes of the liner layer -> spikes of the linear layer
the STAC-ST model also shows to learn the task of time-aligned speaker change detection.-> is shown?

---

> ### Author Rebuttal · Authors · 2023-08-28
>
> Thanks for your time and insightful reviews.
>
> Reason To Reject 1 (the merit of the SL and TL tokens is not reported on):
> - As introduced in Section 3.2, [SL] and [TL] are language tokens used to “define the task for either ST (when [SL]!=[TL]) or ASR (when [SL]==[TL]).” For a joint ST+ASR model, we have to provide these two tokens to specify which task to perform. We do have a ST-only baseline in Row-1 of Table 2, which is equivalent to not using language tokens.
>
> Reason To Reject 2 (comparing table 6 and 5, it seems that Whisper is completely robust):
> 1. BLEU scores in Table 5 and Table 6 are not comparable, because Table 5 uses MT-MS segments (see Section 4.3) while Table 6 uses single-turn segments. Even with the same hypothesis, different segmentations lead to different BLEU scores.
> 2. We can hardly draw any conclusion on the robustness to MT-MS vs. single-turn scenario given existing data points. Whisper has higher BLEU scores on MT-MS segments than on single-turn segments; STAC-ST as higher BLEU scores on single-turn segments than on MT-MS segments.
>
> Reason To Reject 3 (The single speaker results from other papers are 4+ years old; the size of these models is not provided):
> - Table 6 lists most recent papers that report scores on Fisher-CALLHOME, and E2E-ST (Inaguma et al., 2019) is the latest we could find. Whisper, which was trained on massive data and released in 2022, is generally considered a very strong baseline. Only Multi-task (Weiss et al., 2017) reports the model size, which is 9.8M.
> - We compare with more recent models on one additional single-turn test set: CoVoST2. In the following table, we report BLEU scores on 3 translation directions (German/French/Spanish→English) and compare with 3 recent papers that report BLEU scores on CoVoST2: Whisper, XLS-R, and CoVoST2. We list our STAC-ST models ranging from S-size to L-size. They were trained on CoVoST2 ST and Common Voice ASR data with both single-turn and synthetic multi-turn segmentations as introduced in Section 4.2. The Fisher-CALLHOME training data was also used for the Spanish→English model. Whisper, XLS-R and CoVoST2 A2E-M are multilingual models. For fair comparison, we trained a multilingual STAC-ST L-size model by combining data of all related languages. Our languages tokens specify the translation direction.
> - |Model                               |Multilingual?|Size|DE→EN|FR→EN|ES→EN|
> |------------------------------------|:-----------:|---:|----:|----:|----:|
> |Whisper-base (Radford et al., 2022) |      Y      | 74M| 11.7| 15.4| 21.3|
> |Whisper-small (Radford et al., 2022)|      Y      |244M| 25.3| 27.3| 33.0|
> |XLS-R 0.3B (Babu et al., 2021)      |      Y      |300M| 26.7| 32.9| 34.1|
> |CoVoST2 Bi-ST (Wang et al., 2021)   |      -      |  --| 17.1| 26.3| 23.0|
> |CoVoST2 A2E-M (Wang et al., 2021)   |      Y      |  --| 18.9| 27.0| 28.0|
> |(Ours) STAC-ST S                    |      -      | 21M| 19.7| 29.2| 29.1|
> |(Ours) STAC-ST M                    |      -      | 86M| 20.5| 31.8| 32.6|
> |(Ours) STAC-ST L                    |      -      |298M| 21.4| 25.2| 33.0|
> |(Ours) STAC-ST L - Multilingual     |      Y      |298M| **27.5**| **34.0**| **35.8**|
> - The results show that (1) our multilingual large model outperforms Whisper and XLS-R multilingual models with comparable sizes, even though Whisper and XLS-R where trained on data two orders of magnitude larger: 680k hours for Whisper, 436k hours for XLS-R, and ~3k hours for STAC-ST L multilingual; (2) our models with smaller sizes sometimes outperform larger Whisper models, such as STAC-ST 21M vs. Whisper 244M on French→English. These results along with our original paper demonstrate that our proposed approach is well-suited for both the novel single-channel multi-speaker speech translation task and the conventional pre-segmented speech translation.
>
> Reason To Reject 4 (Shouldn't finetuning Whisper models be the SOTA baseline?):
> - Fine-tuning Whisper models on Fisher-CALLHOME data is an interesting idea, but it raises many technical questions. For example, should we preserve their original multi-task objectives, including voice activity detection, language identification, and ASR? Should we fine-tune the model with our proposed objectives? The latter would be an interesting extension to leverage pre-trained models for improvements, which is considered as future work.
>
> Reason To Reject 5 (Please confirm only using ground-truth for the "chunking" of the test data for processing):
> - You are right, we only use ground-truth segmentation information to chunk the test data for pre-processing, and we do not provide ground-truth turn annotations to the model during inference. We will make it more clear in the paper.

---

### Official Review · Reviewer_RZ8H · 2023-08-11

**Soundness:** 4

**Excitement:**

4: Strong: This paper deepens the understanding of some phenomenon or lowers the barriers to an existing research direction.

**Paper Topic And Main Contributions:**

This paper tackles single-channel multi-speaker conversational speech translation with an end-to-end and multi-task training model, named Speaker-Turn Aware Conversational Speech Translation (stac-st). It combines automatic speech recognition, speech translation and speaker turn detection using special tokens in a serialized labeling format.
Experiments on Fisher-callhome corpus show that this paper's proposed method can achieve good results.
Authors said that they will release the data processing and model training code.

**Questions For The Authors:**

Questions:
1. can you show some exmples of speech translation that are benefit from stac-st? Prefer to learn the cases of phonemes whose translation results are improved by the rich context-based stac-st architecture.

**Reasons To Accept:**

Strong:
1. multiple speaker conversational speech translation is an impressive direction and this paper is meaningful in practice.
2. stac-st achieved sotat results with detailed ablation study.

**Reasons To Reject:**

weak:
1. besides fisher/callhome datasets, do we also have results in terms of traditional speech translation datasets?
2. the paper will be evaluated higher if the baseline models can be enriched by including more recent st models after 2019, besides Whisper model.

**Reproducibility:**

4: Could mostly reproduce the results, but there may be some variation because of sample variance or minor variations in their interpretation of the protocol or method.

**Reviewer Confidence:**

4: Quite sure. I tried to check the important points carefully. It's unlikely, though conceivable, that I missed something that should affect my ratings.

---

> ### Author Rebuttal · Authors · 2023-08-28
>
> Thanks for your time and insightful reviews.
>
> Reason To Reject 1 (results of traditional speech translation datasets):
> - Traditional speech translation datasets are composed of single-turn pre-segmented utterances. Following Section 5.3.3 “STAC-ST for Single-Turn ST”, we also run experiments on the CoVoST2 test set. In the following table, we report BLEU scores on 3 translation directions (German/French/Spanish→English) and compare with 3 recent papers that report BLEU scores on CoVoST2: Whisper, XLS-R, and CoVoST2. We list our STAC-ST models ranging from S-size to L-size. They were trained on CoVoST2 ST and Common Voice ASR data with both single-turn and synthetic multi-turn segmentations as introduced in Section 4.2. The Fisher-CALLHOME training data was also used for the Spanish→English model. Whisper, XLS-R and CoVoST2 A2E-M are multilingual models. For fair comparison, we trained a multilingual STAC-ST L-size model by combining data of all related languages. Our languages tokens specify the translation direction.
> - |Model                               |Multilingual?|Size|DE→EN|FR→EN|ES→EN|
> |------------------------------------|:-----------:|---:|----:|----:|----:|
> |Whisper-base (Radford et al., 2022) |      Y      | 74M| 11.7| 15.4| 21.3|
> |Whisper-small (Radford et al., 2022)|      Y      |244M| 25.3| 27.3| 33.0|
> |XLS-R 0.3B (Babu et al., 2021)      |      Y      |300M| 26.7| 32.9| 34.1|
> |CoVoST2 Bi-ST (Wang et al., 2021)   |      -      |  --| 17.1| 26.3| 23.0|
> |CoVoST2 A2E-M (Wang et al., 2021)   |      Y      |  --| 18.9| 27.0| 28.0|
> |(Ours) STAC-ST S                    |      -      | 21M| 19.7| 29.2| 29.1|
> |(Ours) STAC-ST M                    |      -      | 86M| 20.5| 31.8| 32.6|
> |(Ours) STAC-ST L                    |      -      |298M| 21.4| 25.2| 33.0|
> |(Ours) STAC-ST L - Multilingual     |      Y      |298M| **27.5**| **34.0**| **35.8**|
> - The results show that (1) our multilingual large model outperforms Whisper and XLS-R multilingual models with comparable sizes, even though Whisper and XLS-R where trained on data two orders of magnitude larger: 680k hours for Whisper, 436k hours for XLS-R, and ~3k hours for STAC-ST L multilingual; (2) our models with smaller sizes sometimes outperform larger Whisper models, such as STAC-ST 21M vs. Whisper 244M on French→English. These results along with our original paper demonstrate that our proposed approach is well-suited for both the novel single-channel multi-speaker speech translation task and the conventional pre-segmented speech translation.
>
> Reason To Reject 2 (more recent st models after 2019):
> - Table 6 lists most recent papers that report scores on Fisher-CALLHOME, and E2E-ST (Inaguma et al., 2019) is the latest we could find. Whisper, which was trained on massive data and released in 2022, is generally considered a very strong baseline. We include more recent models on CoVoST2 above.
>
> Question (examples of speech translation that are benefit from stac-st):
> - We provide translations with and without using STAC-ST here.
> - |System   |Translation|
> |---------|-----------|
> |Reference|... hello good evening **who is this [turn] [xt] how's it going** hey this is guillermo ...|
> |Baseline |... hello good evening **how are you** i'm guillermo ...|
> |STAC-ST  |... hello good evening **who is it [turn] [xt] how is it going** eh i'm guillermo ...|
> - In this example, the second speaker jumps in while the first speaker is saying “who is this”. The baseline model trained with only single-turn data fails to handle the cross-talk and cuts off “who is this”. Our STAC-ST model not only accurately identifies the speaker-turn change and cross-talk (by producing [turn] [xt]), but also successfully serializes the cross-talk.
> - Note that we remove all special task tokens before quantitative evaluation as explained in Section 4.3.

---

### Official Review · Reviewer_BkF5 · 2023-08-12

**Soundness:** 3

**Excitement:**

3: Ambivalent: It has merits (e.g., it reports state-of-the-art results, the idea is nice), but there are key weaknesses (e.g., it describes incremental work), and it can significantly benefit from another round of revision. However, I won't object to accepting it if my co-reviewers champion it.

**Paper Topic And Main Contributions:**

This work is to address the multi-turn and multi-speaker challenges in conversational speech translation. The paper demonstrates experimental results appealing the proposed end-to-end speaker-turn aware conversational speech translation (STAC-ST), which involves serialized multi-task labeling framework based on special tokens.

**Reasons To Accept:**

- The motivation is clear, and the proposed approach sounds technical.
- The experimental results show encouraging improvement.

**Reasons To Reject:**

- The explanation about the challenge is not detailed. I'd like to see more illustration or even analysis, maybe quantative analysis, about the multi-turn and multi-speaker problem, and a comprehensive study about relationship of the proposed approach and the improvement. After reading the whole paper, I feel the relationship of the proposed method and the task is not tight. If the task is not speech translation, it seems the serialized labeling based on task tokens may still work.
- The findings seem not novel. Joint training of single-turn and multi-turn, ST and ASR, and multi-turn ASR can improve the performance of multi-turn ST sound intuitive.

**Reproducibility:**

3: Could reproduce the results with some difficulty. The settings of parameters are underspecified or subjectively determined; the training/evaluation data are not widely available.

**Reviewer Confidence:**

4: Quite sure. I tried to check the important points carefully. It's unlikely, though conceivable, that I missed something that should affect my ratings.

---

> ### Author Rebuttal · Authors · 2023-08-28
>
> Thanks for your time and insightful reviews.
>
> Reason To Reject 1 (I feel the relationship of the proposed method and the task is not tight.):
> - Our ablation study has already provided comprehensive **quantitive** analysis on how each component of our approach contributes to the improvement of multi-turn multi-speaker (MT-MS) ST. Section 5.1 shows that standard training with single-turn data only achieves 50-60% BLEU of our approach. Section 5.2 shows that text-labeling speaker-turn and cross-talk enables speaker change detection at acoustic level, which yields ~1.5 BLEU improvement. We will make these more evident in Section 5.1 and 5.2.
> - We are not sure the meaning of “If the task is not speech translation, it seems the serialized labeling based on task tokens may still work.” But we would like to comment that: (1) We use the generic idea of “serialized labeling based on task tokens” for our specific task of MT-MS ST by injecting speaker-turn and cross-talk information. (2) “Serialized labeling” is just one component of our approach and evaluation. We experimented and analyzed many other factors to the MT-MS problem.
>
> Reason To Reject 2 (The findings seem not novel. Joint training of ... sounds intuitive):
> - Our key contribution is not simply using multi-task learning, but exploring an under-studied yet realistic problem: single-channel multi-turn & multi-speaker ST. When tackling the problem, we also explored speaker-turn and cross-talk detection without providing time-aligned signals.

---

### Meta-Review · Area_Chair_9xF7 · 2023-09-08

**Recommendation:** 5

**Metareview:**

I concur with the other reviewers and believe this paper's idea is valuable; I recommend acceptance.

---

### Decision · Program_Chairs · 2023-10-07

**Decision:**

Accept-Main

**Comment:**

I concur with the other reviewers and believe this paper's idea is valuable; I recommend acceptance.